

# Founder effects drive the genetic structure of passively dispersed aquatic invertebrates

Javier Montero-Pau[1,2], Africa Gómez[2] and Manuel Serra[3]

[1] Department of Biochemistry and Molecular Biology, Universidad de Valencia, Valencia, Spain
[2] Department of Biological Sciences, University of Hull, Hull, United Kingdom
[3] Instituto Cavanilles de Biodiversidad y Biología Evolutiva, Universidad de Valencia, Valencia, Spain

## ABSTRACT

Populations of passively dispersed organisms in continental aquatic habitats typically show high levels of neutral genetic differentiation despite their high dispersal capabilities. Several evolutionary factors, including founder events, local adaptation, and life cycle features such as high population growth rates and the presence of propagule banks, have been proposed to be responsible for this paradox. Here, we have modeled the colonization process to assess the impact of migration rate, population growth rate, population size, local adaptation and life-cycle features on the population genetic structure in these organisms. Our simulations show that the strongest effect on population structure are persistent founder effects, resulting from the interaction of a few population founders, high population growth rates, large population sizes and the presence of diapausing egg banks. In contrast, the role of local adaptation, genetic hitchhiking and migration is limited to small populations in these organisms. Our results indicate that local adaptation could have different impact on genetic structure in different groups of zooplankters.

# INTRODUCTION

Successful dispersal and colonization are essential for the establishment and persistence of species, and an understanding of these processes is crucial in the face of changing climate, habitat loss, agricultural intensification and biological invasions, which are rapidly affecting the abundance and distribution patterns of many species (*Parmesan & Yohe, 2003*; *Dawson et al., 2011*; *Chen et al., 2011*; *Blackburn, Lockwood & Cassey, 2015*; *Hallmann et al., 2017*). The evolutionary outcome of dispersal and colonization results from a potentially complex interplay of neutral and non-neutral factors, founder effects or bottlenecks causing genetic drift during the first stages of colonization, inbreeding depression, or high levels of gene flow that could erode local adaptation (*Lenormand, 2002*; *Kliber & Eckert, 2005*; *Rosenblum, Hickerson & Moritz, 2007*; *Keller & Taylor, 2008*; *Verhoeven et al., 2011*). In addition, life-cycle features and demographic characteristics may act as modulators and lead to different evolutionary outcomes (*Burton, Phillips & Travis, 2010*). For example, after a bottleneck, species with high population growth rates

Corresponding author
Javier Montero-Pau,
javier.montero@uv.es

are more likely to maintain their genetic variability ("founder-flush" model) (*Carson, 1968*; *Templeton, 2008*), and populations of organisms with resistant life stages (e.g., diapausing eggs) are more likely to be connected by migration even at long distances (*Frisch, Green & Figuerola, 2007*). Predicting the impact of of neutral and selective factors and life history features on population structure is a major question in evolutionary and conservation ecology, and requires an understanding of the effect of each factor and their interactions.

Populations of passively dispersed aquatic invertebrates (e.g., freshwater bryozoans, rotifers, cladocerans, copepods, anostracans, notostracans) and macrophytes inhabiting lentic habitats typically present a high level of neutral genetic differentiation (*Freeland, Romualdi & Okamura, 2000*; *Zierold, Hanfling & Gómez, 2007*; *Mills, Lunt & Gómez, 2007*; *Muñoz et al., 2008*; *Makino & Tanabe, 2009*; *Xu et al., 2009*; *Escudero et al., 2010*), despite their high dispersal capabilities through diapausing propagules (*Allen, 2007*; *Frisch, Green & Figuerola, 2007*; *Vanschoenwinkel et al., 2011*) has been termed the "migration-gene flow paradox". Regardless of their taxonomic disparity, these organisms share biological features promoting a rapid monopolization of resources in the new environment: high population growth rates, large population sizes and the production of resistant stages in their life cycle. Resistant stages can accumulate in sediments and form dormant propagule banks (*Hairston, 1996*; *Brendonck & De Meester, 2003*) and constitute the dispersal stage. As a result, once a habitat becomes available and is colonized, the population can grow very quickly creating a numerical advantage that dilutes the genetic impact of further migrants (*Waters, Fraser & Hewitt, 2013*), resulting in a persistent founder effect (*Boileau, Hebert & Schwartz, 1992*). This explanation was expanded by *De Meester et al. (2002)* into the so called "Monopolization Hypothesis" (MH hereafter) to include local adaptation as an important force contributing to reduce effective gene flow and therefore maintaining the genetic structure of passively dispersed aquatic organisms. The MH postulates that the migration-gene flow paradox could be explained by a combination of three factors: (1) persistent founder effects, (2) selection against migrants due to local adaptation and (3) buildup of linkage disequilibrium between neutral markers and genes under selection.

Local adaptation is an important and rapid process in many zooplanktonic organisms (*Cousyn et al., 2001*; *Decaestecker et al., 2007*; *Costanzo & Taylor, 2010*; *Declerck et al., 2015*; *Tarazona, García-Roger & Carmona, 2017*). The impact of local adaptation on population genetic structure is diverse as it is dependent on the impact of other evolutionary forces, not only selection (*Kawecki & Ebert, 2004*). For instance, it can promote genetic differentiation, 'isolation-by-adaptation' sensu (*Nosil, 2007*) or reinforce the existing genetic differentiation by reducing effective gene flow (*Orsini et al., 2013*). Irrespective of local adaptation, populations recently founded by a small number of propagules can be highly inbred and show inbreeding depression (*De Meester, 1993*; *Tortajada, Carmona & Serra, 2009*) could give migrants a fitness advantage and favor gene flow into the population (*Ebert et al., 2002*; *Haag et al., 2006*). The accumulation of large numbers of resistant stages (i.e., propagule banks) in sediments is also a characteristic of many aquatic species inhabiting temporary habitats. These banks have an important role in community-level ecological (*Chesson, 1983*; *Cáceres, 1997*; *Montero-Pau & Serra, 2011*) and evolutionary processes (*Brendonck & De Meester, 2003*). They increase the effective population size due to postponed reproduction

in the bank, and thus reduce genetic drift (*Kaj, Krone & Lascoux, 2001*). However, this effect may be indirect, as gene flow is also postponed in the propagule bank (*Kaj, Krone & Lascoux, 2001*; *Berg, 2005*).

The relative importance and interactions between the demographic features of organisms and the neutral and selective processes acting during colonization has remained poorly understood. Therefore, an explicit analysis of the effects of local adaptation, persistent founder effects, and their interplay on the differentiation of populations of aquatic organisms is due, especially during the first stages of colonization when populations are still small and, thus, more sensitive to stochastic effects.

Here, we have modeled the colonization process of zooplanktonic organisms to clarify how migration rate, growth rate, population size, local adaptation and the existence of a propagule bank shape the population genetic structure during the first stages of colonization. Our primary interest is to gain insights into the relative importance of (1) persistent founder effects, (2) selection against migrants as a consequence of local adaptation, and (3) random associations between neutral genes and genes under selection (linkage disequilibrium). Our simulation results show that persistent founder effects have the strongest effects on population structure, resulting from the interaction of a few population founders, high population growth rates, large population sizes and the presence of diapausing egg banks. In contrast, local adaptation, genetic hitchhiking and migration have impacts on the population structure only in small populations.

## MATERIALS AND METHODS

We developed a genetic and demographic model to analyze the effects of population growth rate, population size, presence of a diapausing egg bank and local adaptation on the population genetic structure of aquatic organisms. We assumed a geographic scenario with two habitats, with local populations connected through reciprocal migration. These local populations are founded simultaneously after a single event of migration from a source population.

The model was based on the life cycle of rotifers and cladocerans (i.e., cyclical parthenogenesis), which are major groups of passively dispersed continental aquatic invertebrates. Cyclical parthenogenesis combines parthenogenesis with episodic sexual reproduction and typically consists of several asexual generations followed by a sexual generation, generally associated with seasonal habitat degradation. The sexual generation produces diapausing eggs that hatch into asexual individuals once the habitat becomes suitable again. As not all eggs hatch from one growth period to the next, they may accumulate in the sediment and form extensive diapausing egg banks (*Brendonck & De Meester, 2003*).

The demographic submodel is outlined in Fig. 1. Briefly, it consists in six steps:

step 1.     Hatching of diapausing eggs (resident and migrant)
step 2.     Asexual proliferation
step 3.     Sexual reproduction and production of diapausing eggs
step 4.     Diapausing eggs survival in the sediment
step 5.     Migration of diapausing eggs
step 6.     Back to step 1.

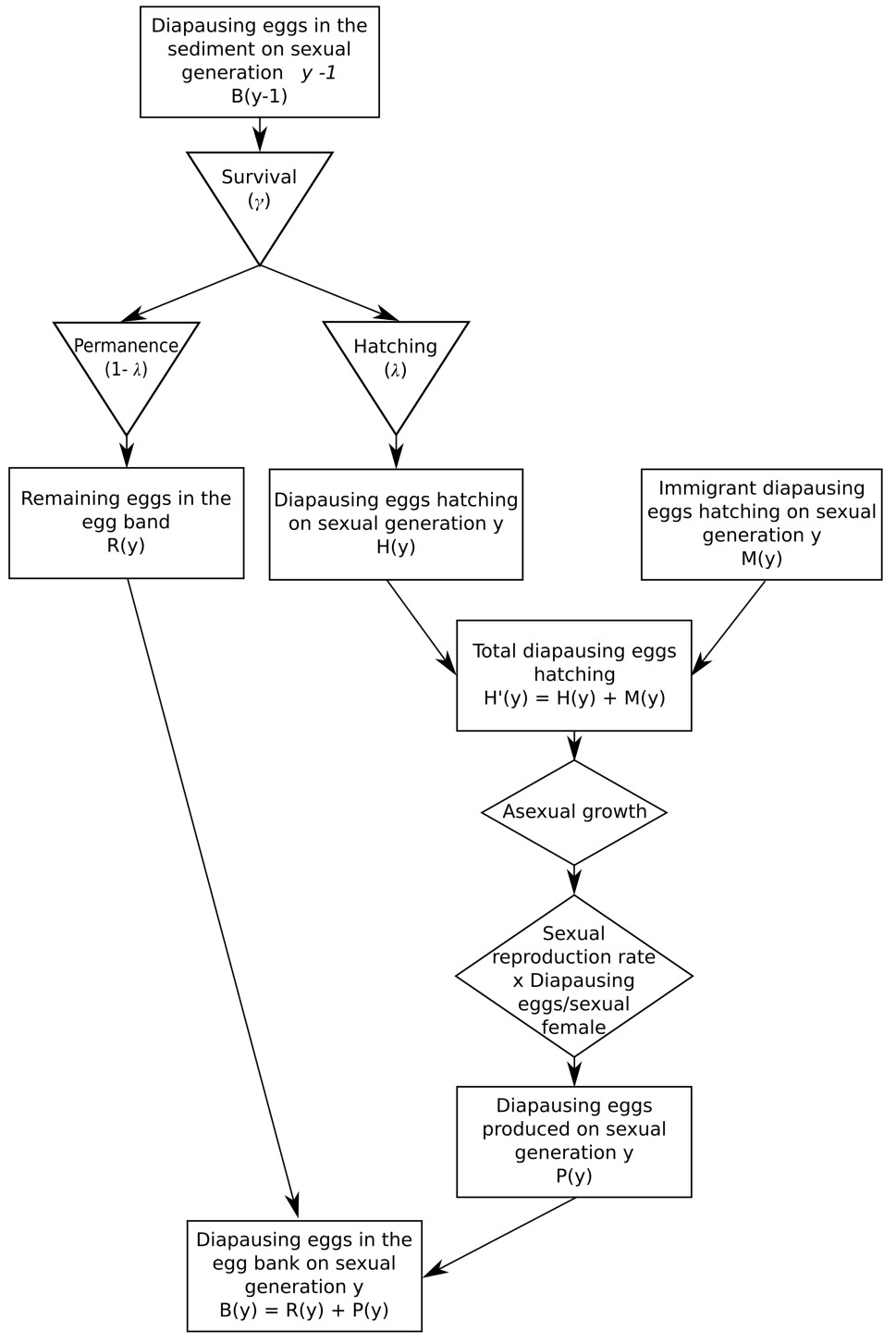

**Figure 1** Demographic submodel.

Note that this demography implies two time scales: (1) a within-planktonic growth period (often within-year; index, t), and (2) an among-sexual generations scale (often among-years; index, y).

Migration, either from the source population or between habitats, is assumed to occur via diapausing eggs, which are passively transferred between habitats, and their hatching time is assumed to be the same as for locally produced diapausing eggs. As migration rates are low relative to the size of diapausing egg bank, migration is assumed to have negligible effects on the source bank.

## Genetic submodel

All individuals are considered to have $n$ neutral loci and $n$ loci under selection. All loci are biallelic and no mutation is assumed. Absence of linkage is assumed among neutral loci and among loci under selection. Contrarily, the model accounts for physical linkage between each neutral locus and a corresponding selected locus, assuming several recombination levels, including absence of linkage. Loci under selection act additively on growth rate. Consequently, no dominance and no epistatic effects are assumed. Local adaptation requires a genotype-environment interaction on fitness, which is modeled through $\delta_{i,j,l}$, which is the effect on the intrinsic growth rate (see below) of allele $i$ ($i$: 1, 2) at locus $j$ ($j$: 1,…, n) in locality $l$ ($l$: 1, 2). The assumptions are (1) $\delta_{1,j,1} = \delta{2,j,2}$, and (2) $\delta_{i,j,l} = -\delta_{j \neq i,j,l}$; so, in the case of homozygotes for a given selected locus, they will experience an increase or decrease of their growth rate by $|2\delta|$ depending on the locality. Hence, the growth rate for each genotype $g$ in each locality $l$ ($r_{g,l}$) can be decomposed into $r$ (basal growth rate) and $\theta$ (deviation of each genotype), so that

$$r_{g,l} = r + \theta_{g,l}$$

where $g$ is the genotype, $l$ is the locality, and $\theta_{g,l}$ is the summation of the fitness components ($\delta$) in locality $l$ of the alleles carried by a genotype $g$ in the $n$ loci under selection. Thus, in any given locality, the growth rate during the asexual reproduction will vary between the limits $r \pm 2n\delta$.

Sexual reproduction is assumed to be panmictic and, for simplicity, is considered to be synchronic and at the end of the growing season ($t = \tau$). As linkage disequilibrium can occur due to selection and genetic drift, gametic frequencies are computed. Gametes are then drawn to produce the diapausing eggs.

Genetic distance between populations was estimated based on neutral loci as

$$F_{\mathrm{ST}} = \frac{\overline{H_T} - \overline{H_S}}{\overline{H_T}},$$

where $\overline{H_T}$ is the average expected heterozygosity for the two populations considered as a single one for the neutral loci, and $\overline{H_S}$ the average of the mean expected heterozygosity within each populations for the neutral loci (*Hedrick, 2011*). Allelic frequencies for each locus were computed using the total number of alleles. Similarly, a genetic distance for loci under selection ($F_{\mathrm{STQ}}$) was computed (*Le Corre & Kremer, 2012*). $F_{\mathrm{ST}}$ and $F_{\mathrm{STQ}}$ values were obtained just after hatching of diapausing eggs.

## Population growth

The asexual phase spans from time $t = 0$ to $\tau$, which is the moment when sexual reproduction takes place. During the asexual phase the population grows deterministically

according to a logistic growth model:

$$\frac{dN_{l,g}}{dt} = N_{l,g}r_{l,g}\left[1 - \frac{\Sigma_g N_{l,g}}{k}\right]$$

where $N_{l,g}$ is the density of the genotype $g$ in the locality $i$, $r_{l,g}$ is its intrinsic population growth rate during the asexual phase, and $K$ the carrying capacity. Note that $K$ is genotype-independent. At the onset of each asexual growth season ($t = 0$), $N_{l,g}$ is the sum of the hatched diapausing eggs, a fraction of them having been locally produced $H_{l,g}$, and the rest being migrants $M_{l,g}$.

At $t = \tau$ of the sexual generation $y$, the number of diapausing eggs produced $P_{l,g}$ (y) is stochastically computed from $N_{l,g}(\tau, y)$ assuming a sexual proportion $m$ (fraction of the females that becomes sexual), a sex ratio $sr$ and an effective fecundity $e$ (number of diapausing eggs produced per sexual female).

Mortality of diapausing eggs in the sediment with egg bank was assumed to be age-independent (annual survival rate $\gamma$). Empirical information supporting this assumption for field populations is not available. However, our model can account for fast senescence when it assumes the absence of egg bank. When a new planktonic growing season starts ($t = 0$) a fraction $\lambda$ of the diapausing eggs in the sediment hatches.

## Source population and local population founding

The two populations are founded at time $y = t = 0$ by $F$ diapausing eggs randomly drawn from a single source population. The source population is assumed to be in Hardy-Weinberg equilibrium and of infinite size, so that extraction of migrants does not change genotype frequencies. All loci are considered neutral in the source population, so no preadaptation to any of the populations exists.

## Model implementation

The impact of carrying capacity ($K$), growth rate ($r$), migration ($M$), selection pressure ($\delta$) and recombination rate on $F_{ST}$'s were analyzed by exploring a range of realistic values for zooplanktonic organisms. $K$ was varied from $2 \times 10^2$ to $2 \times 10^7$ individuals, which is equivalent to densities from 0.001 to 100 individuals/L in a small pond of 200 m$^2$ and 1 m depth, in good agreement with reported average densities of cladocerans and rotifers (*Carmona, Gómez & Serra, 1995*; *Ortells, Gómez & Serra, 2003*; *Tavernini, 2008*). $r$ was explored from 0.05 to 1 days$^{-1}$. Cladocerans show maximum $r$ of 0.2–0.6 days$^{-1}$ and rotifers 0.2–1.5 days$^{-1}$ (*Allan, 1976*). The number of population founders ($F$) was set to 1 diapausing egg across most simulations. That is, foundation is considered a rare event. Note that as the model assumes cyclical parthenogenesis, a single diapausing egg is enough for population establishment. Studies on passively dispersed aquatic invertebrates show that the number of founders is low (*Haag et al., 2005*; *Louette et al., 2007*; *Ortells, Olmo & Armengol, 2011*; *Badosa et al., 2017*). For example, *Haag et al. (2005)* reported and average of 1.7–1.8 colonizers for two species of *Daphnia*, and 57% of studied populations were likely founded by a single individual. The effect of the numbers of founders ($F$) was also explored (1, 2, 5, 50 diapausing eggs). Other parameter values used in the simulations are shown in Table 1.

**Table 1  Summary of model parameters and assumed values.**

| Parameter | Definition | Value |
|---|---|---|
| $F$ | Number of founders (individuals) | 1–50 |
| $M$ | Number of immigrants per sexual generation (individuals) | $0$–$10^5$ |
| $\gamma$ | Annual survival proportion of eggs in the egg bank | 0.763[a] |
| $\lambda$ | Annual hatching proportion of diapausing eggs | 0.046[a] |
| $y$ | Sexual generations | 1,000/4,000 |
| $\tau$ | Duration of the asexual growth period (days) | 60 |
| $r$ | Clonal growth rate of each genotype (days$^{-1}$) | 0.05–1.00 |
| $K$ | Carrying capacity (individuals) | $2 \times 10^2 - 2 \times 10^7$ |
| $m$ | Sexual proportion | 0.7[b] |
| $sr$ | Sex ratio | 0.5[c] |
| $e$ | Diapausing egg production per sexual female | 3 |
| $n$ | Number of neutral loci | 5 |
| $\delta$ | Additive value on $r$ (days $^{-1}$) | $10^{-5}$–$10^{-1}$ |

**Notes.**
[a]Calculated from *García-Roger, Carmona & Serra (2006b)*.
[b]*Alver & Hagiwara (2007)*.
[c]*Aparici, Carmona & Serra (1998)*.

Simulations considered two scenarios regarding diapausing egg banks: (1) an annual, age-independent, diapausing egg survival rate on the sediment ($\gamma = 0.763$) (i.e., a diapausing egg bank); and (2) $\gamma = 0.763$ for eggs of age = 1 and a $\gamma = 0$ for older eggs (i.e., absence of diapausing egg bank). Parameters for the diapausing egg bank ($\gamma$ and $\lambda$, the annual hatching rate) were estimated from rotifer diapausing egg banks (*García-Roger, Carmona & Serra, 2006b*) by adjusting them to the model described by *García-Roger, Carmona & Serra (2006a)*.

The simulation model was implemented in C++ and based on Monte-Carlo procedures (code available at https://github.com/monpau/founder_effects). The Mersenne twister algorithm (*Matsumoto & Nishimura, 1998*) was used as random number generator. The logistic model was iterated numerically. 50 replicates for each parameter combination (but 100 for values of $\delta$ and recombination rate) were performed. For each replicate, a source population was randomly created by drawing from a uniform distribution the allelic frequencies of the n and s loci. After foundation of the two populations, 1,000 sexual generations (4,000 generations for some scenarios) were simulated.

Sampling effects were taken into account for hatching and survival of diapausing eggs if the total number of eggs in the population was lower than 1,000. Selection of migrants and gametes for mating were performed randomly regardless of the number of eggs/individuals involved.

Paralleling the procedure in an empirical study, a statistical assessment was performed. Differences between $F_{ST}$ values under a neutral scenario and scenarios with selective pressure and different recombination rates were analysed with an ANOVA and *a priori* contrasts. Correlations between $F_{ST}$ and $F_{STQ}$ at different combinations of population size, recombination rates and selective pressure were also tested using Kendall's Tau and

Sperman's Rho. All statistical analyses were performed using SPSS v. 17 (SPSS Inc., Chicago, USA).

## RESULTS

The population dynamics of a newly founded population, using the parameters shown in Table 1, with $\tau = 60$ days and $r = 0.3$ days$^{-1}$—which are realistic values for both the length of the growth season (*Tavernini, 2008*) and the intrinsic growth rate of many aquatic invertebrates (*Allan, 1976*) –show that carrying capacity ($K$) is reached in less than two sexual generations, even in the case of the highest $K$ (i.e., $K = 2 \times 10^7$ individuals). Thus, $K$ is a good proxy of population size and we will use both terms interchangeably hereafter.

### Effect of migration

The effect of the number of migrants on genetic differentiation of neutral loci strongly depends on $K$ (i.e., population size; Fig. 2). In both the small and the large populations, $F_{ST}$ decreases with increasing migration rates, as expected under a neutral scenario (*Wright, 1931*). For the lowest carrying capacity tested ($K = 2 \times 10^2$ individuals; Fig. 2A), $F_{ST}$ decreased rapidly down to very low levels with increasing migration. In contrast, for the highest $K$ tested ($K = 2 \times 10^7$ individuals; Fig. 2B), $F_{ST}$ was rather insensitive to the effect of migration, and populations remained highly differentiated ($F_{ST} > 0.2$) even at high levels of migration, as predicted by persistent founder effects. Note that the high level of genetic differentiation observed is due to the strong bottlenecks during colonisation (see below and Fig. S1). The number of migrants needed to cause a noticeable decrease of genetic differentiation on neutral loci is in the order of 100 and 1,000 individuals/sexual generation for the situation without and with diapausing eggs respectively.

### Effect of population size

Carrying capacity (i.e population size) had strong effects on $F_{ST}$ (Fig. 3; 1,000 sexual generations). In small populations (i.e., low $K$) populations did not differ genetically, while in large populations, $F_{ST}$ remained as high as the values observed just after population foundation, reflecting the importance of migration and persistent founder effects respectively. In the case of the low carrying capacity scenario (low $K$), migration seems to be stronger than the effect of genetic drift and a reduction of genetic differentiation is observed. At higher $K$, genetic differentiation can only operate during the earliest generations before $K$ is reached. When $K$ is very high, high population densities are achieved very quickly and genetic differentiation remains equal to the genetic differentiation after founding (i.e., persistent founder effect). At intermediate $K$ high population sizes are not achieved as soon, so genetic drift can operate during a few more generations. In other words, the highest $F_{ST}$ values are found at intermediate population sizes. The $F_{ST}$-$K$ pattern is qualitatively similar with and without diapausing egg bank, but in absence of a egg bank a lower maximum $F_{ST}$ at a higher $K$ was found.

   These results are robust to changes in the maximum number of sexual generations explored (results for maximum $y = 100, 500, 2,000$ and $4,000$ generations, data not shown). However, at 100 and, to a lesser extent, 500 sexual generations, the peak of $F_{ST}$

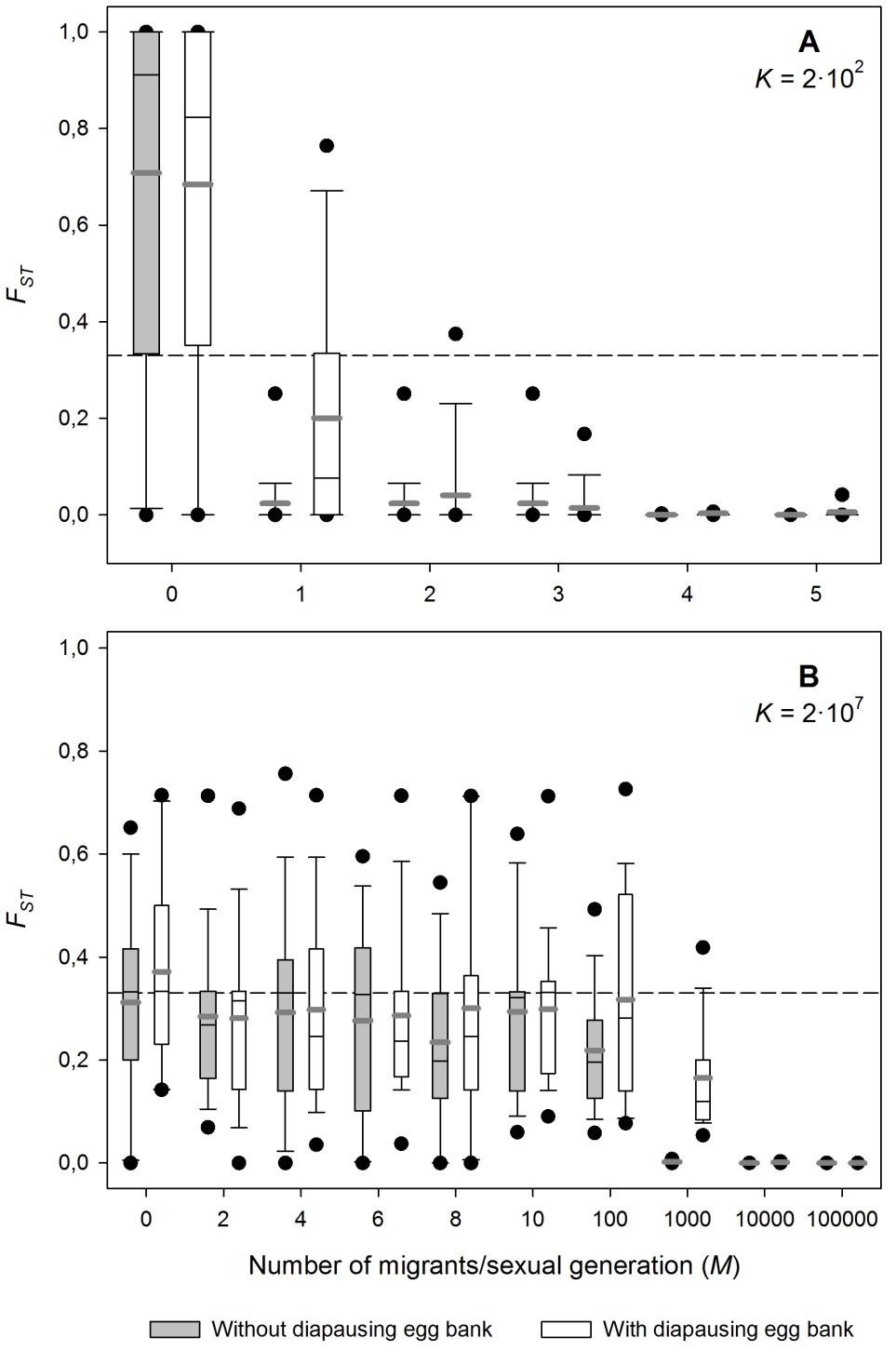

Without diapausing egg bank    With diapausing egg bank

**Figure 2** **Effect of migration ($M$) on population differentiation ($F_{ST}$) after 1,000 sexual generations with and without a diapausing egg bank.** Population differentiation ($F_{ST}$) after 1,000 sexual generations plotted against migration ($M$) with and without a diapausing egg bank for (A) $K = 2 \times 10^2$, and (B) $K = 2 \times 10^7$ individuals. The rest of parameters were $r = 0.3 \, d^{-1}$, $n = 5$, $s = 0$ and $F = 1$. Box plots are based on 50 replicate simulations. Boxes represent 25th /75th percentile and black dots the 5th/95th percentile. Thin black lines and thick gray lines in each bar represent the median and the mean, respectively. Dashed horizontal lines show the initial value of $F_{ST}$ after foundation.

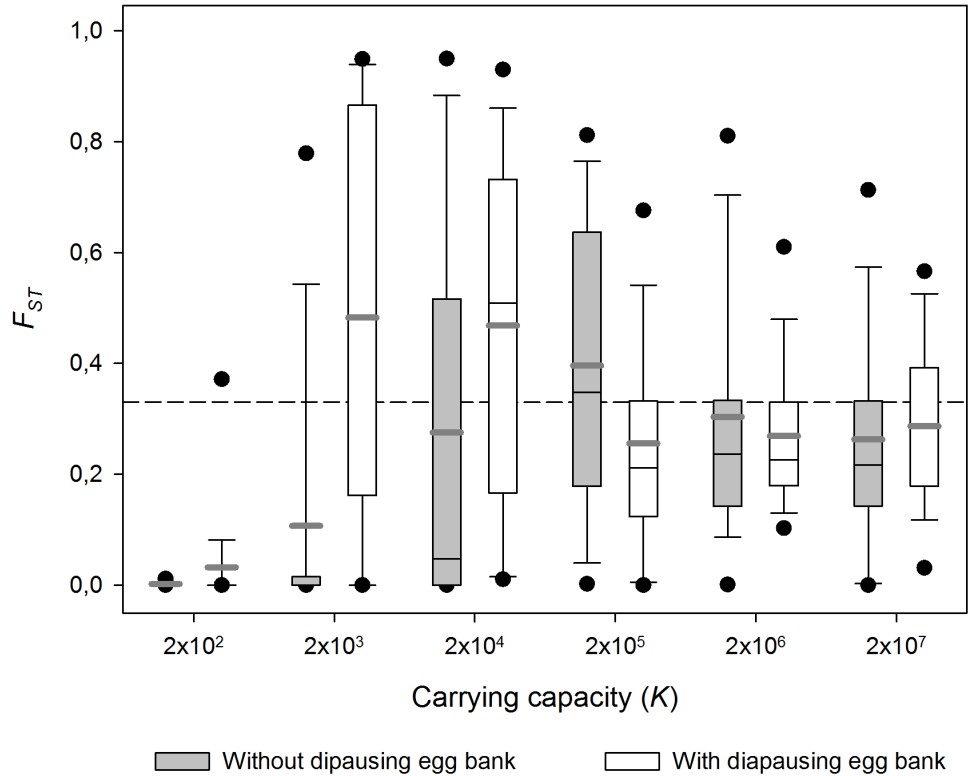

**Figure 3  Effect of carrying capacity ($K$) on population differentiation ($F_{ST}$) with and without a diapausing egg bank.** Population differentiation ($F_{ST}$) after 1,000 sexual generations plotted against carrying capacity ($K$) with and without a diapausing egg bank. Simulation values for other parameters were $r = 0.3\ d^{-1}$, $n = 5$, $s = 0$, $F = 1$ and $M = 2$. Data is based on 50 replicate simulations. Boxes represent 25th /75th percentile and black dots the 5th/95th percentile. Thin black lines and thick gray lines in each bar represent the median and the mean, respectively. Dashed horizontal line shows the initial value of $F_{ST}$ after foundation.

at intermediate population sizes was less pronounced than at later sexual generations. The long-term (from 1st to the 4,000th sexual generation) time course is further explored in Fig. 4. In the absence of a diapausing egg bank (Fig. 4A), $F_{ST}$ decreases with time at low population size, and the situation is reversed when $K$ increases, to finally become virtually constant (i.e., determined by the initial condition) at the largest population size explored ($K = 2 \times 10^7$). A qualitatively similar pattern is found when a diapausing egg bank is present (Fig. 4B), although the shift to an increasing $F_{ST}$ time course, and also to $F_{ST}$ constancy, occurs at lower population sizes. Note that the small negative change found at $K = 2 \times 10^2$ (regardless whether a bank is assumed) is associated to the very low initial $F_{ST}$ values (Figs. 4C, 4D). Also note that $F_{ST}$ values are calculated after hatching of residents and migrants; for instance, at $y = 1$, $F_{ST}$ value is not the value after foundation but after migration.

In summary, population size and presence or absence of a diapausing egg bank are key to predict the main force shaping the genetic structure. Decreasing $F_{ST}$ indicates that migration is becoming the dominant factor, while increasing values show that drift
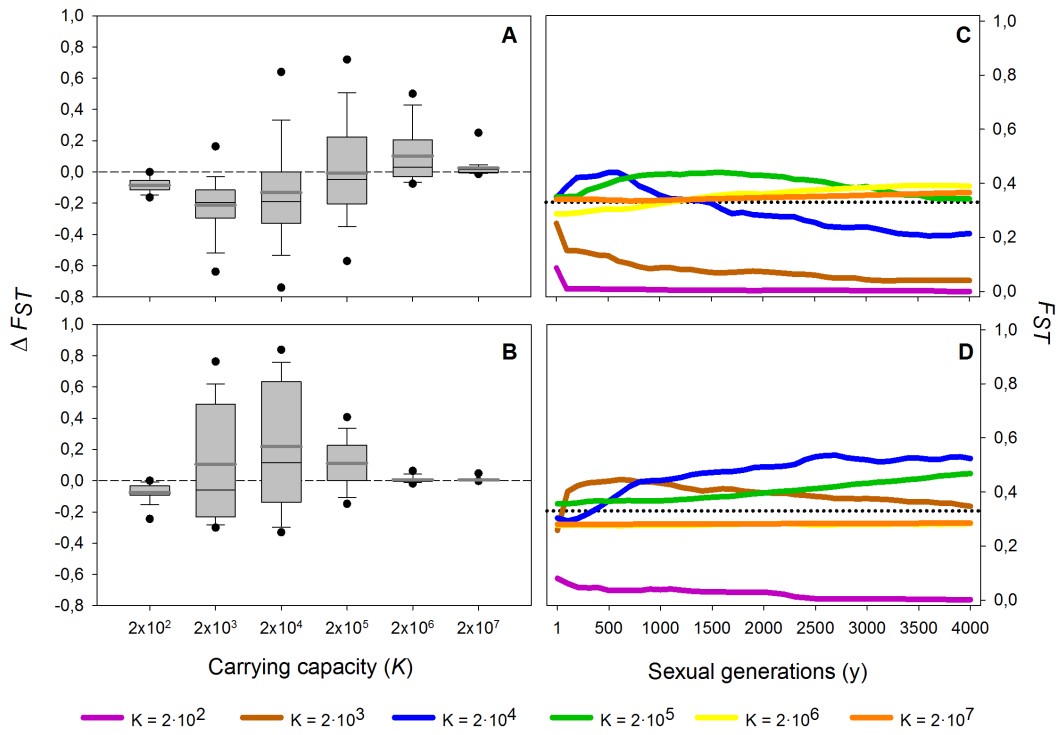

**Figure 4** **Effect of different carrying capacities ($K$) on $F_{ST}$ along 4,000 sexual generations.** Effect of different carrying capacities ($K$) on $F_{ST}$ along 4,000 sexual generations. (A, B) Box plot of the increment of $F_{ST}$ ($\Delta F_{ST}$) after 4,000 sexual generations (A) without and (B) with diapausing egg bank is shown. (C, D) Time course of the average $F_{ST}$ values along 4,000 generations (C) without and (D) with diapausing egg bank. Simulation conditions were $r = 0.3\ d^{-1}$, $n = 5$, $s = 0$, $F = 1$ individual and $M = 2$ individuals. Data is based on 50 replicates. Boxes represent 25th /75th percentile and black dots the 5th/95th percentile. Thin black lines and thick gray lines in each bar represent the median and the mean, respectively. Dashed horizontal lines show the initial value of $F_{ST}$ after foundation.

becomes dominant. Our finding of practically stationary $F_{ST}$ from the first generations after foundation indicates the importance of persistent founder effects on the shaping of the genetic structure of populations.

Population growth rate interacts with population size in determining the level of genetic differentiation (Fig. 5). Low growth rates result in low genetic differentiation, regardless of population size, indicating a high impact of migration. However, for population growth rates above $0.1\ d^{-1}$, which are common for zooplanktonic organisms, genetic differentiation becomes sensitive to variations in population size.

## Effects of the number of founders

As expected, increasing the number of population founders $F$ results in a dramatic decrease of $F_{ST}$ values just after foundation (Fig. 6); for instance, if compared to $F = 1$, $F_{ST}$ is reduced by half for $F = 2$, and approaches 0 for $F = 50$. This effect is persistent, as after 4,000 sexual generations, the level of population differentiation still shows a negative relationship with the number of founders. Given the strong effect of the number of founders, we explored in further simulations how $F$ affects the relationships between population differentiation

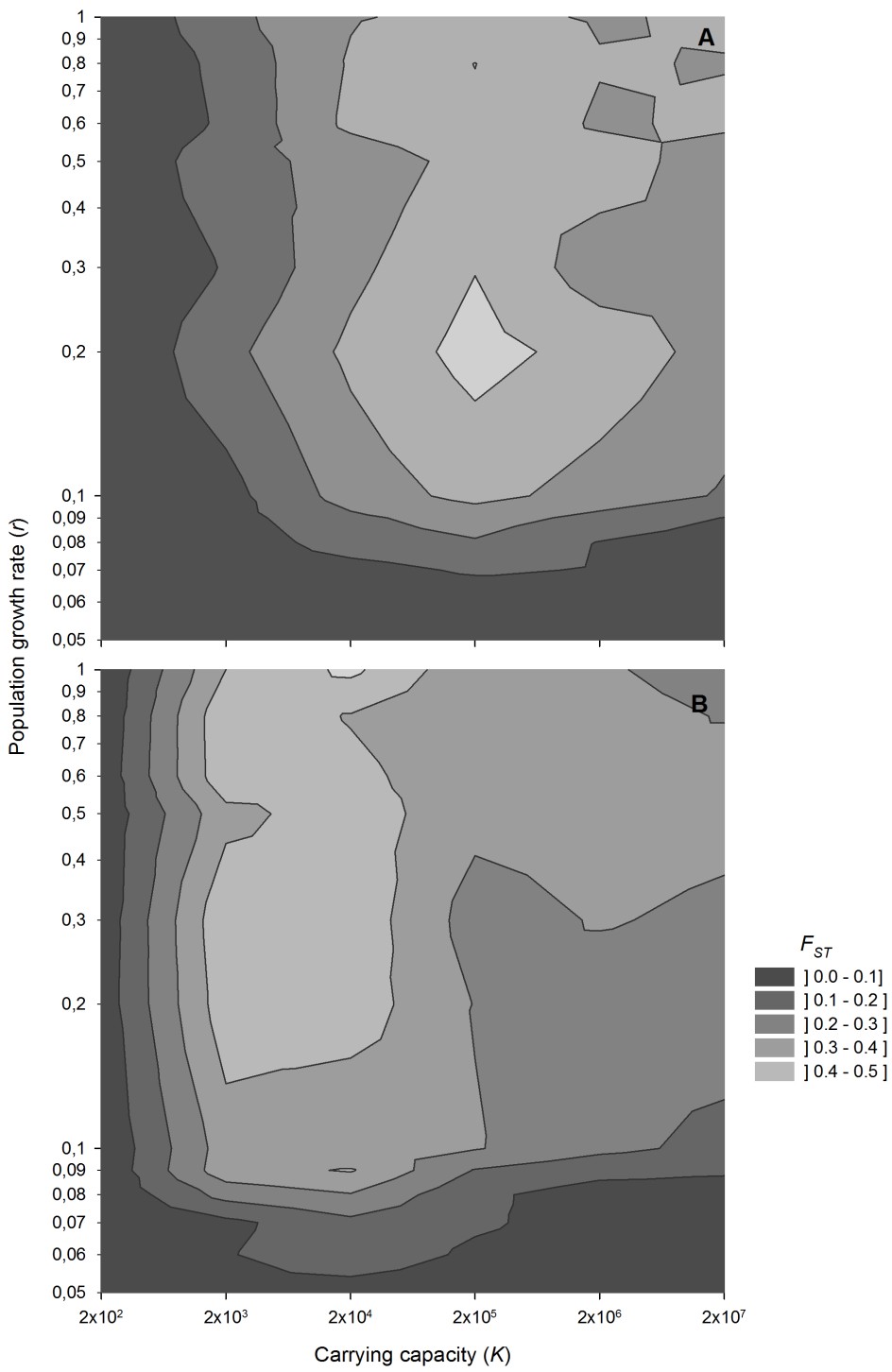

**Figure 5  $F_{ST}$ values after 1,000 sexual generations at different combinations of population growth rates and carrying capacity.** Contour plot showing $F_{ST}$ values after 1,000 sexual generations at different combinations of population growth rates and carrying capacity (A) without and (B) with diapausing egg bank. Simulation conditions were $n = 5$, $s = 0$, $F = 1$ and $M = 2$. Data is based on 50 replicates.

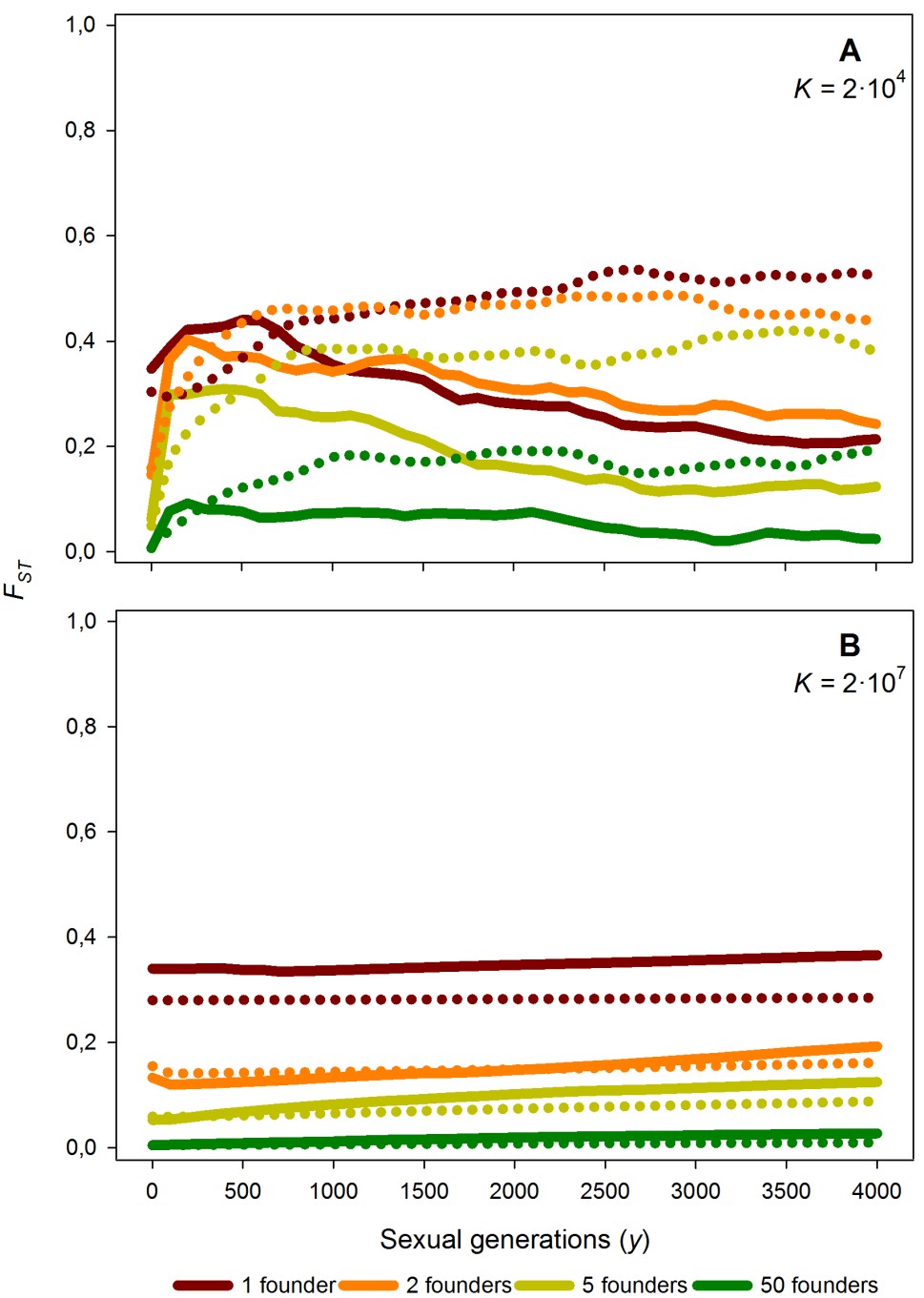

**Figure 6  Average $F_{ST}$ value along 4,000 generations for different number of founders and carrying capacity.** Time course of the average $F_{ST}$ value along 4,000 generations for different number of founders ($F = 1, 2, 5$ and $50$), for $K = 2 \times 10^4$ (A) and $K = 2 \times 10^7$ (B) and $M = 2$. Solid lines: without diapausing egg bank, dotted lines: with diapausing egg bank. Average $F_{ST}$ values obtained from 50 replicates.

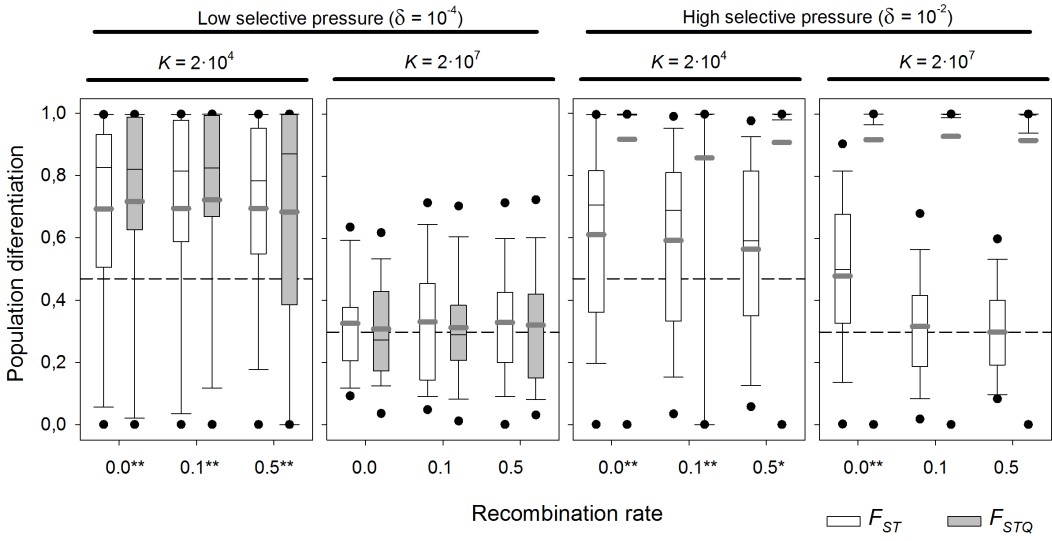

**Figure 7 Population differentiation ($F_{ST}$ and $F_{STQ}$) after 1,000 sexual generations with different recombination rates at two different selection scenarios.** Box plot graph of $F_{ST}$ and $F_{STQ}$ values after 1,000 sexual generations with different recombination rates for two different values of fitness components ($\delta = 10^{-4}$ and $10^{-2}$ $d^{-1}$) and with presence of a diapausing egg bank. For each of the fitness scenario, the left panel refers to $K = 2 \times 10^4$ and the right panel to $K = 2 \times 10^7$. Other parameters were $r = 0.3\ d^{-1}$, $n = 5$, $s = 5$, $F = 1$ and $M = 2$. Data is based on 100 replicates. Boxes represent 25th/75th percentile and black dots the 5th/95th percentile. Thin black lines and thick gray lines in each bar represent the median and the mean, respectively. Dashed horizontal lines show the initial value of $F_{ST}$ after foundation. Asterisks indicate $F_{ST}$ statistically different from those without selection ($\delta = 0$) (**, $\alpha = 0.05$; *, $\alpha = 0.1$).

and other factors. Our results suggest that the patterns outlined above are qualitatively maintained for $F > 1$ (data not shown and Fig. S1).

## Effect of local adaptation

Above, a selectively neutral scenario was assumed. The effect of local adaptation was explored at two levels of $K$ ($2 \times 10^4$ and $2 \times 10^7$ individuals), which are realistic values for cladocerans and rotifers respectively. Two different selection scenarios ($\delta = 10^{-4}$ days$^{-1}$, weak selection, and $10^{-2}$ days$^{-1}$, strong selection) in the presence/absence of diapausing egg bank, and six recombination rates—from complete linkage to unlinked genes—were tested (Fig. 7 summarizes the results for the scenario with diapausing egg bank; see Fig. S2, for the equivalent scenario without diapausing egg bank).

With strong selection, $F_{STQ}$ reaches almost maximum values—i.e., populations are almost fixed for the locally adapted alleles—regardless of $K$ (Fig. 7). In the case of low $K$, all $F_{ST}$ values are statistically different from those obtained without selection ($p$-values $<0.05$ except at 0.5 recombination rate; $p$-value $= 0.057$). However, $F_{ST}$ values are similar irrespective of the recombination rate. In contrast, for high $K$, only those values of $F_{ST}$ with complete linkage (recombination rate $= 0$) are statistically different of those found without selection. This indicates that genetic hitchhiking in large populations acts only on neutral loci tightly linked to those under selection. Otherwise, linkage to the genes under selection does not reduce the persistence of founder effects.

With weak selection, $F_{STQ}$ indicates the expected result that local adaptation becomes less important than with strong selection. In large populations ($K = 2 \times 10^7$), $F_{ST}$ values do not statistically differ from the neutral scenario, showing the larger importance of founder effects over local adaptation when selection is weak. Moreover, $F_{STQ}$ values also appear to be affected by persistent founder effects. In contrast to the situation with strong selection, genetic linkage does not alter differentiation at neutral loci. However, in small populations ($K = 2 \times 10^4$), local adaptation does play a role. Mean $F_{ST}$ values statistically differ from the neutral scenario at all recombination rates (from 0.0 to 0.5), and the variance of the distribution of $F_{ST}$ values is decreased (see Fig. 3 for comparison). Note that drift is the dominant factor in relatively small ($K = 2 \times 10^4$) populations with diapausing egg bank.

### Effects of diapausing egg banks

In the presence of strong selection the effects of diapausing egg bank (see Supplementary Fig. S2) were minimal. In weak selection conditions: (1) at high population density ($K = 2 \times 10^7$) genes under selection are less affected by persistent founder effects that when no bank is present (Fig. 7), and populations show a trend to be locally adapted; (2) at low population density ($K = 22 \times 10^4$), $F_{ST}$ values at recombination rates 0.0 and 0.1 are statistically different from the neutral scenario—unlike at higher recombination rates—, which indicates that genetic hitchhiking could be of some importance; (3) at $K = 2 \times 10^4$ $F_{ST}$ and $F_{STQ}$ had higher variance at all recombination rates than in the scenario with no diapausing egg bank (Fig. 7).

In the absence of a diapausing egg bank, populations reach maximum $F_{STQ}$ values in about 40–50 sexual generations regardless of population size (data not shown). However, when a diapausing egg bank exists, advantageous alleles need a longer time to reach fixation (about 150 sexual generations for $K = 2 \times 10^4$, and about 300 generations for $K = 2 \times 10^7$).

We computed $F_{STQ}$ vs. $F_{ST}$ correlations within each tested parameter combination. Significant correlations were found only in the case of the low $K$ ($22 \times 10^4$) without diapausing egg bank. Correlation coefficient is always positive, and the ranges are: Kendall's tau = 0.66–0.53 and Spearman's rho = 0.73–0.56 for strong selection; Kendall's tau = 0.68–0.32 and Spearman's rho = 0.80–0.38 for weak selection.

## DISCUSSION

The understanding of the evolutionary factors responsible for the strong population structure of passively dispersed aquatic organisms in the face of potentially high gene flow has attracted considerable attention (*De Meester et al., 2002*; *Mills, Lunt & Gómez, 2007*; *Campillo et al., 2009*). We have presented a specific model and, by simulation, explored the effects of genetic drift associated to population founding (founder effects), gene flow via migration and local adaptation on genetic differentiation. Our results show that the strongest effect were persistent founder effects, resulting largely from the distinctive life history traits of these organisms: few population founders, high rates of population growth, large population sizes and the presence of diapausing egg banks. These results are in agreement with those of Boileau et al. (*Boileau, Hebert & Schwartz, 1992*), who proposed that persistent founder effects are an important force shaping the genetic

structure of passively dispersed aquatic organisms. A novel aspect of our findings is that the persistent founder effects hold in a scenarios with selection and genetic linkage. The most remarkable and novel result of our simulations is that the role of local adaptation and genetic hitchhiking on shaping genetic structure of these organisms is not significant in large populations, although it plays a significant role in small populations. Recent data in the rotifer *B. plicatilis* in fact indicates very low levels of genomic linkage, meaning that the opportunities for hitchhiking are limited and selection can freely act in very specific regions of the genome without affecting nearby loci (*Franch Gras et al., 2018*).

In agreement with *Boileau, Hebert & Schwartz (1992)*, migration has a very limited effect on the population structure of passively dispersed aquatic organisms. For instance, a migration rate of 1,000 individuals per sexual generation is needed to cause a noticeable effect on $F_{ST}$ in a large population. Although direct estimates of the number of dispersing stages are unavailable, this extremely large value appears unlikely to occur between non-connected ponds (*Cáceres & Soluk, 2002*; *Allen, 2007*; *Frisch, Green & Figuerola, 2007*), and is inconsistent with estimates of the number of founders in natural populations, which are expected to be correlated with regular migration rates (*Louette et al., 2007*; *Badosa et al., 2017*). However, for small population sizes our model recovers the expected pattern for the combined effect of migration and drift under neutral genetic differentiation.

Among the factors studied in our model, population size was largely responsible for establishing the levels of genetic differentiation observed in natural populations of aquatic organisms. In addition, the effect of population size was strongly reinforced when a diapausing egg bank is established. Although egg banks could increase gene flow (*Kaj, Krone & Lascoux, 2001*; *Berg, 2005*), the dominant effect is to buffer the effects of migration and reducing genetic drift, which favors the establishment of persistent founder effects. In our model, we assumed a parameter range in agreement with values reported for many aquatic organisms (*Montero-Pau, Serra & Gómez, 2017*; *Hairston, 1996*; *Tavernini, 2008*). Nevertheless, due to computational limitations the maximum values used for population sizes and egg bank densities had to be limited, and could underestimate those attained in many natural populations. For example, some estimated population sizes and diapausing egg bank densities in rotifers are one or two orders of magnitude higher than the maximum values considered here (*Carmona, Gómez & Serra, 1995*; *Ortells, Gómez & Serra, 2003*). Diapausing egg bank densities for zooplanktonic organisms are in the order of $10^3$–$10^7$ eggs/m$^2$ (*Hairston, 1996*), although densities in the sediment layers that could provide recruits are uncertain. However, modeling larger population sizes is unlikely to change our results qualitatively; if anything, they would increase the relative impact of persistent founder effects.

Local adaptation seems to be common and has been well documented in cladocerans (*Cousyn et al., 2001*; *De Meester et al., 2002*; *Decaestecker et al., 2007*) and the generalist rotifer *Brachionus plicatilis* (*Campillo et al., 2010*; *Franch-Gras et al., 2017*). However, the effect of local adaptation on genetic structure does not seem to be general, as it is weakened by neutral and demographic factors. Regarding differentiation in genetic markers, a limited role for local adaptation in continental aquatic invertebrates has been suggested (*Campillo et al., 2009*; *Allen, Thum & Cáceres, 2010*). Our results indicate that although

local adaptation does occur, its effects on population structure are only apparent when population sizes and diapausing egg banks are relatively small. Given that rotifers tend to have larger population sizes than cladocerans, we could predict that the effects of local adaptation on population structure could differ between these organisms. According to our results, genetic hitchhiking appears to be of limited importance in shaping neutral genetic differentiation. We have only detected signs of its effect (1) at completely linked genes with high population size and strong selection, and (2) at intermediate population size without egg bank and weak selection. The lack of observed impact does not mean that genetic hitchhiking has no importance, but that other processes are dominating the outcome. We must stress that our main question is not whether local adaptation occurs or not, but itsit effects on genetic differentiation in neutral markers *sensu* Nosil (*Nosil, 2007*). Admittedly, due to computational limitations our model simplifies the selective scenarios acting on continental aquatic invertebrates. As selection in natural populations of aquatic invertebrates is likely to be multifactorial and fluctuating, this scenario should be further explored.

Genetic analyses in recently established populations indicate that the number of founders is small (*Haag et al., 2005*: average of 1.8 founders; *Louette et al., 2007*:1 to 3 founders; *Ortells, Olmo & Armengol, 2011*: 1 to 7 founders; *Badosa et al., 2017*: 1 to 10 founders). Accordingly a single founder was assumed in most simulations. By assuming a single founder, the studied scenario corresponds, for instance, to a situation where a new region consisting of several lakes is open to colonization—e.g., after glaciation—, with few founders of any single lake, but with varying migration rates among lakes. When we relaxed the assumption of a single founder, the only remarkable observed effect was a reduction on the initial value of $F_{ST}$, but the same patterns of genetic differentiation change were observed

Globally, our results show that persistent founder effects, genetic drift and local adaptation all drive population genetic structure in these organisms, but population size and the presence of a diapausing egg bank have a strong control on the dominance of each of these factors. In turn, these demographic variables can be linked to ecological features. If so, a habitat classification linking ecological factors, demographic features, and mechanisms acting on genetic structure could be possible. Therefore, in populations inhabiting permanent ponds and lakes where a low investment in diapause is generally found—as reflected in small diapausing egg banks in comparison to temporary or ephemeral ponds (*Hebert, 1974a*; *Hebert, 1974b*; *García-Roger, Carmona & Serra, 2006b*; *Campillo et al., 2010*; *Montero-Pau, Serra & Gómez, 2017*)—an increased effect of local adaptation and genetic drift is expected. In contrast, in environmental conditions limiting population sizes, such as small rock pools or nutrient-poor lakes, migration can attain higher importance. If despite this, high genetic differentiation is detected, an effect of selective forces can be hypothesized. For instance, genetic hitchhiking has been suggested for a *Daphnia* metapopulation inhabiting temporal rock pools (*Haag et al., 2006*). Besides ecological features, our results suggest that differences can be expected between taxa differing in body size and so in their typical population sizes, and therefore differences between the smaller rotifers and the larger cladocerans are expected. As far as our results

identify a restricted number of factors driving the genetic structure, they provide insights beyond the life cycle assumed (i.e., cyclical parthenogenesis), and could be extended to organisms with similar demographic features (i.e., high growth rates, high population densities or presence of seed or egg banks). For example, populations of sexual species with high growth rates (i.e., *r* strategists) like crustaceans such *Artemia* or copepods, which produce egg banks, are also likely to benefit from a numerical inertia that will reduce the impact of migration on the genetic structure of their populations (*Boileau, Hebert & Schwartz, 1992*).

As we have shown, the rapid growth rate of colonists acts as a barrier against new migrants, which is reinforced by the formation of diapausing stage banks and, in some cases, by local adaptation leading to a persistent founder effect, and consequently, to a deviation from the migration-drift equilibrium. This has repercussions when interpreting phylogeographic signals (*Gómez, Carvalho & Lunt, 2000*; *Waters, 2011*). For instance patterns of 'isolation-by-distance' found in several aquatic organisms, regardless of their reproductive mode, have been suggested to be due to a process of sequential colonizations (*Gouws & Stewart, 2007*; *Gómez et al., 2007*; *Mills, Lunt & Gómez, 2007*; *Muñoz et al., 2008*; *Paz-Vinas et al., 2015*). Our results are consistent with these proposals and suggest that caution should be applied when inferring a migration-drift mechanism of 'isolation by distance' from such patterns (i.e., correlation between genetic and geographical distances). Also, the establishment of persistent founder effects and competitive exclusion of closely related species can explain the phylogenetic overdispersion in communities, given a phylogenetic limiting similarity between species (*Violle et al., 2011*).

During the time window from the arrival of first colonizers to the establishment of the founder effects, the genetic structure of the population is still sensitive to migration or drift. Our results point out that this period is short, as a result of the high population growth rates of most aquatic organisms. Nevertheless, we found that with relatively low population growth rates, the numerical advantage is delayed and genetic differentiation is relatively low. Inbreeding depression is expected to be larger in small populations (*Lohr & Haag, 2015*), and it could act favoring gene flow (*Haag et al., 2002*; *Tortajada, Carmona & Serra, 2009*; *Tortajada, Carmona & Serra, 2010*). Although we did not explicitly model inbreeding depression here, it will act in a similar way of reducing the growth rate, which will favor effective gene flow. However, severe inbreeding could also reduce the effective population size, and increase genetic drift, which will increase genetic differentiation. A more detailed exploration of this scenario will be needed and it will depend on the relative magnitude of the purging and migration.

Other factors not implemented in our model, but likely to occur in the wild, could also counteract the high genetic differentiation. For example, processes able to reduce population size during asexual growth phase (e.g., perturbations or environmental fluctuations) could increase the impact of gene flow. In addition, it will be of interest to test the strength of persistent founder effects buffering migrants with a higher fitness than locally adapted residents. These factors—inbreeding depression, environmental fluctuations, and preadapted migrants—were not invoked in the initial formulation of the Monopolization Hypothesis and should be investigated in future analyses. An additional

prospect is to include the effect of metapopulation structure. *Walser & Haag (2012)* have shown that population turnover, which is expected to have high rate in small populations, could also explain high genetic population differentiation.

### Concluding remarks

Molecular screening of natural population has uncovered an unexpectedly high genetic differentiation in taxa with high dispersal potential. These findings challenged classical views of the evolutionary processes in small multicellular organisms, and when focused on aquatic invertebrates, brought to postulate a combination of processes as causal factors for that genetic differentiation, the Monopolization Hypothesis (*De Meester et al., 2002*). Our analysis shows that a quantitative elaboration of this multifactorial hypothesis is able to dissect the relative weights of the different factors, and their interactions. Specifically, we found that founder effects drive the genetic structure of passively dispersed aquatic organisms. We conclude that although selective factors and migration have a role in explaining genetic structure of continental aquatic invertebrates, demographic processes are dominant. By studying which factors are important in what circumstances, our analysis can help understand relevant differences among the genetic structure of different species.

## ACKNOWLEDGEMENTS

We thank Guillermo García Franco and José Gargallo Tuzón for their invaluable help with some parts of the code and programming support. We also would like to thank Luc De Meester, Raquel Ortells and Ma José Carmona for helpful comments on previous versions of this manuscript.

### Funding

This work was funded by a grant from the Spanish Ministerio de Ciencia e Innovación (CGL2009-07364) to Manuel Serra. Africa Gómez was supported by a National Environment Research Council (NERC) Advanced Fellowship (NE/B501298/1) and Javier Montero-Pau by a fellowship by the Spanish Ministerio de Ciencia y Tecnología (BES2004-5248). The funders had no role in study design, data collection and analysis, decision to publish, or preparation of the manuscript.

### Grant Disclosures

The following grant information was disclosed by the authors:
Spanish Ministerio de Ciencia e Innovación:  CGL2009-07364.
National Environment Research Council (NERC) Advanced Fellowship: NE/B501298/1.
Spanish Ministerio de Ciencia y Tecnología:  BES2004-5248.

### Competing Interests

The authors declare there are no competing interests.

## Author Contributions

- Javier Montero-Pau conceived and designed the experiments, performed the experiments, analyzed the data, prepared figures and/or tables, authored or reviewed drafts of the paper, approved the final draft.
- Africa Gómez and Manuel Serra conceived and designed the experiments, analyzed the data, contributed reagents/materials/analysis tools, authored or reviewed drafts of the paper, approved the final draft.

## Data Availability

Raw data was deposited at GitHub: https://github.com/monpau/founder_effects.

## Supplemental Information

Supplemental information for this article can be found online at http://dx.doi.org/10.7717/peerj.6094#supplemental-information.

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
