# Peer review of "Founder effects drive the genetic structure of passively dispersed aquatic invertebrates"

_PeerJ, doi:10.7717/peerj.6094_

## Round 0.1 · original submission · Minor Revisions

Dear Dr. Montero-Pau and colleagues:

Thanks for submitting your manuscript to PeerJ. I apologize for the lengthy time in review, but we needed more time than usual to find reviewers.

I have now received two independent reviews of your work, and as you will see, both are very favorable. Well done! Nonetheless, the reviewers raised some relatively minor concerns about the research, and areas where the manuscript can be improved. Please note that reviewer 2 has included a marked-up version of the manuscript in his review.

Therefore, I am recommending that you revise your manuscript accordingly, taking into account all of the issues raised by the reviewers. I do believe that your manuscript will be ready for publication once these issues are addressed.

Good luck with your revision,

-joe

Reviewer 1 ·

Basic reporting

Line 56: "Predicting the outcome of these factors…" What are 'these factors'? Are you refering to the two listed in the previous sentence, or the whole set listed in the first paragraph?

LIne 59: "dispersing" The verb tense seems wrong to me. But, this could be a matter of style.

In general, there is an excessive use of ambiguous pronouns to start sentences. Using words like 'it', 'these', 'they', and so on decreases comprehension by the reader. Ambiguous pronouns are used abundantly throughout the MS and I strongly suggest that the authors edit their MS to specifically remove these words, especially when they are used in the begining of a sentence.

Lines 103-108: In this final paragraph of the introduction, the authors outline what they did. It would be nice to follow those enumerated sentences with a few sentences about what they found. What are the main take-aways?

Line 117: "which are major taxonomic groups in the zooplankton". This phrase would imply that zooplankton are themselves a proper taxanomic group which (emphasis on the word 'the'), of course, they are not.

Experimental design

No comment.

Validity of the findings

No comment.

Additional comments

Overall, I found this paper to be interesting and well thought out. I do not have any substantial comments on it and I look forward to seeing it published. I believe that it will be useful for the field.

Reviewer 2 ·

Basic reporting

The MS is clear and well documented. Figures are easily understandable and of good quality.
The MS could nonetheless be checked further by an English native as some sentences read awkward to me (I'm not an English native, I can hence be wrong).

Experimental design

The question is extremely well posed and is relevant. The authors used a simulation-approach to tackle the question and the simulation design is very good to me.
Notably, the set of parameters is well defined and well argumented. My only worry concern the use of a single founding individual (although other scenarios are simulated); to me authors should justify it better, and/or include further results for other scenarios. For instance, authors could easily confront results from 1 vs. 20 founder for some figures to clearly show what the impact of increasing the number of founders is. They argue that 1 founder is a likely scenario for instance for glacial lakes, but there are many other scenarios where I guess 20 founders is also likely.

Validity of the findings

No Comment

Additional comments

That's a good and interesting paper. I have easily read it, it is easy to follow and it tackle a fascinating question. Although I'm not a specialist, the methodology and overall simulation design seem robust to me (but see my comment about the scenario of a single founder).
I have provided some comments and some corrections in an attached file. Two of these comments should be seriously considered by the authors: one concern some results that sounds illogical to me (but authors may have an explanation, and/or I may have missed something) and the other concerns the choice of presenting only the results with a single founder.
I'm sure you'll be able to easily deal with these comments.
Good luck

Annotated reviews are not available for download in order to protect the identity of reviewers who chose to remain anonymous.

---

## Round 0.2 · accepted · Accept

Dear Dr. Montero-Pau and colleagues:

Thanks for re-submitting your manuscript to PeerJ, and for addressing the concerns raised by the reviewers. I now believe that your manuscript is suitable for publication. Congratulations! I look forward to seeing this work in print, and I anticipate it being an important resource for ecologists and evolutionary geneticists studying aquatic invertebrates. Thanks again for choosing PeerJ to publish such important work.

-joe

#

---

## Author Rebuttal · Round 0.2

Dear Editor,

we appreciate very much both reviewers constructive comments. We have revised our manuscript accordingly. We have paid special attention to the comments of Reviewer 2 regarding the use of a limited number of founder and the interpretation of some results. We have included a new figure as he/she suggested exploring the effect of the number of founders and we have improved the description of our results in order to avoid any misinterpretation.

We answer the reviewers comments and detail the changes in the manuscript below. The reviewers comments are in italics and blue, and our answer in regular font.

We hope that the manuscript is now acceptable for publication in PeerJ.

Javier Montero-Pau on behalf of all authors

Answers to Reviewer 1

Line 56: *"Predicting the outcome of these factors…" What are 'these factors'? Are you refering to the two listed in the previous sentence, or the whole set listed in the first paragraph?*

We have rewritten the sentence and the factors are now clearly listed

LIne 59: *"dispersing" The verb tense seems wrong to me. But, this could be a matter of style.*

We have changed this to 'dispersed'.

*In general, there is an excessive use of ambiguous pronouns to start sentences. Using words like 'it', 'these', 'they', and so on decreases comprehension by the reader. Ambiguous pronouns are used abundantly throughout the MS and I strongly suggest that the authors edit their MS to specifically remove these words, especially when they are used in the begining of a sentence.*

We have revised the manuscript to remove ambiguity, improve the grammar and increase clarity throughout.

Lines 103-108: *In this final paragraph of the introduction, the authors outline what they did. It would be nice to follow those enumerated sentences with a few sentences about what they found. What are the main take-aways?*

We have outlined the main results at the end of the introduction. Reviewer 2 suggested to include specific hypothesis, but we have opted for incorporating only the main results, not to make this final paragraph in the introduction too long.

Line 117: *"which are major taxonomic groups in the zooplankton". This phrase would imply that zooplankton are themselves a proper taxonomic group which (emphasis on the word 'the'), of course, they are not.*

We have removed 'taxonomic' from this sentence and rewritten the sentence for clarity.

Answers to Reviewer 2

Reviewer 2 raises two main concerns:

1) one based on the interpretation of some results regarding the effect of the carrying capacity:

- Lines 259: *Drift is supposed to be high at low K. I really don't understand how is it possible. Can you please be more precise in your explanation why Fst is higher at intermediate values, and not at low values of K as normally expected?*

Genetic drift operates at low population levels. In the case of the low carrying capacity scenario (low $K$), migration seems to be stronger than the effect of genetic drift and a reduction of genetic differentiation is observed. At higher $K$, genetic differentiation can only operate during the earliest generations before $K$ is reached. When $K$ is very high, high population densities are achieved very quickly and genetic differentiation remains equal to the genetic differentiation after founding (i.e. persistent founder effect). At intermediate $K$ high population sizes are not achieved as soon, so genetic drift can operate during a few more generations.

We have included a clarification on the manuscript regarding this comment.

- Lines 248-250: *This result sounds rather unlogical for me. With high K value we expect drift not to occur and hence the two large populations to exhibit similar allele frequencies (and hence low Fst value). I guess this is because F = 1 ? Can you please clarify it and provide the same graph with F = 20 ?*

We agree with the reviewer, the high genetic differentiation observed at high $K$ is not due to ongoing genetic drift but to the effect of the establishment of persistent founder effects. Thus, observed $F_{ST}$ is similar to the one obtained after colonization, which is in turn affected by the number of founders.

To avoid any misunderstanding, we have included the clarification that persistent founder effects are responsible of the levels of genetic differentiation at high $K$, and we also have included the graph that the reviewer suggested showing the effect of changing the number of founders.

2) The other concern is related to the use of a a single founder:

- *My only worry concern the use of a single founding individual (although other scenarios are simulated); to me authors should justify it better, and/or include further results for other scenarios. For instance, authors could easily confront results from 1 vs. 20 founder for some figures to clearly show what the impact of increasing the number of founders is. They argue that 1 founder is a likely scenario for instance for glacial lakes, but there are many other scenarios where I guess 20 founders is also likely.*
- Line 368: *I might have missed something, but you did not provide numbers in the article. To which extent you simulations are realistic regarding the number of founders in natural systems? Can you discuss this specific point as it is central to the model and the results.*
- Line 410: *Yes but is it as small as a single founder? It seems to me that this scenario is extreme and it may not conform to most natural situations. To me you should consider further the results of simulations with more than one founder. Or justify further that a single founder is the norm rather than the exception*.

Studies on passively dispersed aquatic invertebrates show that the number of founders is usually low. For example, Haag et al. 2005 report and average of 1.7 - 1.8 colonizers per pond for two species of *Daphnia* (range of 1 to 9 colonizers), and 57% of studied populations were likely founded by a single individual. Louette et al. 2007 also found an average of 1.7 colonizers. Ortells et al. 2011 and Badosa et al. 2017 report values of colonizers among 1 to 10. Thus our decision of using 1 founder lies within the usually observed range of founders and is a sound value for many natural situations. We have introduced a sentence providing further justification for this parameter (line 208).

We agree with the reviewer that the number of founders can have an impact on the levels of genetic differentiation. In fact, we discuss on the manuscript the effect of this parameter on $F_{ST}$ by exploring a range of values. However, our results show that the main variation is the initial level of $F_{ST}$ after foundation, but the dynamics of change of genetic differentiation remains qualitatively similar and the same conclusions regarding the impact of migration, founder effects and genetic drift are obtained. We have made this clearer in the new version (line 294) and we have included a new graph (Supplementary Fig. 1) exploring the effect of the number of founders under different migration rates.

3) Additional comments included in the annotated file:

- Line 97 (*"This sentence is unclear. Please re-write it."*): We have rewritten the sentence for clarity
- Line 106 (*"Could you introduce some predictions regarding these 3 points? This may clarify your thoughts for naïve readers"*): Both reviewers have different suggestions regarding this last paragraph. We have opted for incorporating only the main results, as reviewer 1 suggested, and not the predictions in order not to make this final paragraph in the introduction too long (See answer to Reviewer 1 above).
- As suggested, we have replace the terms "light" and "intense" selection by "weak" and "strong" selection respectively throughout the text.
- Some typos and misspellings (e.g., line 230, line 352) have been corrected.
- Line 376 ("*please rephrase"): We have rephased the sentence*
- Line 447 (*"this is consistent with a recent paper on riverine organisms showing that colonization is the most likely process explaining genetic structuring in rivers. See Paz-Vinas et al. 2015, Mol Ecol"*): Citation has been included.